# Chronic Pediatric Headache as a Manifestation of Shunt Over-Drainage and Slit Ventricle Syndrome in Patients Harboring a Cerebrospinal Fluid Diversion System: A Narrative Literature Review

**DOI:** 10.3390/children11050596

**Published:** 2024-05-15

**Authors:** Dimitrios Panagopoulos, Maro Gavra, Efstathios Boviatsis, Stefanos Korfias, Marios Themistocleous

**Affiliations:** 1Neurosurgical Department, Pediatric Hospital of Athens, 45701 Athens, Greece; mthemistocleous@gmail.com; 2Neuro-Radiology Department, Pediatric Hospital of Athens, 45701 Athens, Greece; mmgavra@yahoo.com; 32nd University Neurosurgical Department, Medical School, General Hospital of Athens ‘Attikon’, University of Athens, 12462 Athens, Greece; eboviatsis@gmail.com; 41st University Neurosurgical Department, Medical School, General Hospital of Athens ‘Evangelismos’, University of Athens, 10676 Athens, Greece; skorfias@otenet.gr

**Keywords:** over-drainage, slit ventricle syndrome, anti-siphon device, programmable valve

## Abstract

The main subject of the current review is a specific subtype of headache, which is related to shunt over-drainage and slit ventricle syndrome, in pediatric patients harboring an implanted shunt device for the management of hydrocephalus. This clinical entity, along with its impairment regarding the quality of life of the affected individuals, is generally underestimated. This is partly due to the absence of universally agreed-upon diagnostic criteria, as well as due to a misunderstanding of the interactions among the implicated pathophysiological mechanisms. A lot of attempts have been performed to propose an integrative model, aiming at the determination of all the offending mechanisms of the shunt over-drainage syndrome, as well as the determination of all the clinical characteristics and related symptomatology that accompany these secondary headaches. This subcategory of headache, named postural dependent headache, can be associated with nausea, vomiting, and/or radiological signs of slim ventricles and/or subdural collections. The ultimate goal of our review is to draw clinicians’ attention, especially that of those that are managing pediatric patients with permanent, long-standing, ventriculoperitoneal, or, less commonly, ventriculoatrial shunts. We attempted to elucidate all clinical and neurological characteristics that are inherently related to this type of headache, as well as to highlight the current management options. This specific subgroup of patients may eventually suffer from severe, intractable headaches, which may negatively impair their quality of daily living. In the absence of any other clinical condition that could be incriminated as the cause of the headache, shunt over-drainage should not be overlooked. On the contrary, it should be seriously taken into consideration, and its management should be added to the therapeutic armamentarium of such cases, which are difficult to be handled.

## 1. Introduction

A recently published meta-analysis, centered on the epidemiological features of pediatric primary headache, estimated that the approximate incidence of migraine in the pediatric and adolescent population overall is in the area of 11% [1]. According to published series, the incidence of pediatric headache varies among the different subgroup of patients, based on their age. Namely, it is more pronounced in children aged approximately 13 years of age [2]. It is worthwhile to mention that headaches are subdivided as primary—that is, of unknown etiology—and secondary, which are intimately related to a relevant pathophysiologic substrate [1]. Among these subtypes of headaches, the headache that accompanies shunt over-drainage and slit ventricle syndrome deserves special mention. The exact frequency of shunt over-drainage postural headache in the pediatric population is unknown as it is difficult to quantify an entity that entails only quality characteristics.

Another parameter that needs to be underlined is that the treatment options are restricted as the pathophysiology that accompanies this spectrum of disorders is not fully elucidated. The main reason for this confusion comes from the limited number of relevant, detailed epidemiological surveys dedicated to the prevalence and incidence of primary headaches in the pediatric age group [1,2]. Moreover, the existing ones are frequently heterogeneous, and this is an intimate characteristic of the intrinsic characteristics of the studies [2,3]. These include age range, sex, social and economic background, the utilized methodologies (e.g., school-based questionnaires, clinician interviews, phone surveys), along with the different inclusion criteria applied, which occasionally could not be considered specific to developmental age [3]. So, when a comparison is attempted with headaches in their adult counterparts, especially due to all of these restrictions, a limited number of epidemiological studies are available in children and adolescents. Namely, based on bibliographic data, the estimated prevalence of headache and migraine is up to 58% and 7.7% [4], respectively. In children and adolescents, their quality of life is substantially impaired by headaches, causing negative feedback in their daily living [5], i.e., the elimination of their social activities and physical activity, school absenteeism, weaker learning outcomes, a higher risk of dropping out of school, and a negative effect on parent’s careers [5,6].

The main purpose of this review is to analyze a subcategory of pediatric headaches that arise as a secondary effect to the iatrogenic management of pediatric hydrocephalus. We have collected relevant data regarding chronic shunt over-drainage and slit ventricle syndrome and have tried to investigate their pathophysiological association with the development of secondary headaches, which are often refractory to medical treatment.

## 2. Materials and Methods

### Search Strategy

We executed a title-specific search using PubMed as well as the Thomson Reuters Web of Science database to identify the articles (reviews, case reports, original research, technical notes) that were related to shunt over-drainage, slit ventricle syndrome, and headache with respect to ependymomas and other posterior fossa tumors (as these patients frequently harbor a ventriculoperitoneal shunt). The time range of our search was extended from 1968—when, to the best of our knowledge, the first bibliographic report on shunt over-drainage appeared—to March 2024. A specific age range was included as a selection criterion; more precisely, our search included only data that were extracted from patients under 18 years of age. Afterwards, we reviewed the results in order to clarify that they were relevant for the purposes of our research. The papers that were chosen were further analyzed in order to extract any conclusions regarding the existence of any association between shunt over-drainage and slit ventricle syndrome and a headache that is resistant to all conservative treatment modalities.

## 3. Discussion

### 3.1. Shunt Over-Drainage and Its Association with Headache

The term shunt over-drainage is utilized in order to delimit a well-known complication that is causally related to excessive drainage of cerebrospinal fluid in patients harboring a CSF shunt system. The term “over-drainage” was first utilized in bibliographic series in 1968 [7,8] and has been increasingly accepted and adopted since the 1990s [9,10]. It is widely known that over-drainage represents one of the most frequently encountered complications that is secondary to CSF shunting procedures [11]. It may be associated with all types of CSF diversion procedures and is not restricted to any specific pediatric age-group but is most-commonly encountered with valve-bearing shunt systems [12]. The clinical equivalent of this pathology is named postural headache and is manifested radiologically with a slender ventricular system (“slit ventricle syndrome”). Due to the common coexistence of these two entities, postural headache is currently being considered as a clinical observation that is frequently recorded in combination with over-drainage [13,14,15,16]. Shunt over-drainage could be combined with different clinical and radiological features, such as subdural hygroma [13,17,18,19] and premature closure of cranial sutures (in infants) [9,20], as well as low ICP syndrome [21]. All of these manifestations should be integrated under the umbrella of shunt over-drainage. There is a wide discrepancy regarding the estimated prevalence of the precise incidence of over-drainage according to current literature data, as it varies from 2 to 71%. The most accepted explanation for this marked fluctuation regarding the statements for this syndrome could be based upon the non-well-specified diagnostic criteria, the heterogeneity of the investigated populations, and the different policies for follow-up after shunting [15,16,22]. Moreover, it is also corroborated that over-drainage may be under-reported, and thus underestimated, due to the lack of consensus regarding the definition criteria of this entity, as well as due to an incomplete knowledge of pathophysiology [16,23]. Consistent with our statements is a survey that was executed among American pediatric neurosurgeons, which documents the lack of consent and the existing uncertainty regarding the understanding and management of over-drainage-associated complications [22]. Apart from that, the range of normal reference values regarding ICP and probably postural CSF pressure/volume regulation seem to be intimately related to age. All of these data imply that the risk of over-drainage, its clinical manifestations, treatment modalities, and protocols may not follow the same pattern in pediatric and adult cohorts. Moreover, this may also be true when toddlers and young teenagers are under investigation [24,25,26,27]. The majority of researchers agree that the remarkable variability regarding the estimated incidence that is referred to in published data is intimately related to the lack of a widely accepted definition [24,25,26,27]. This ultimately results in the absence a of clinical consensus and doubtfulness about diagnosis.

### 3.2. Evolution of Concepts and Current Pitfalls in Shunt Over-Drainage Syndrome

There are several premature—even sparse—previously reported bibliographic reports of inappropriate over-drainage of cerebrospinal fluid in the form of anecdotal cases [8,16,28,29,30,31,32,33]. Fox and coworkers [34] were the first who attempted to record ICP monitoring data in shunted patients. Their data were extracted from 18 patients suffering from normal pressure hydrocephalus; their relevant mean cerebrospinal fluid pressure values were about −220 mm H_2_O for ventriculoperitoneal shunts and about −190 mm H_2_O for ventriculoarterial shunts when they were assuming an upright position. These findings were initially attributed to the siphoning effect of shunts. The initial management option that was adopted was the incorporation of higher-pressure valves, along with VAS, especially for patients who are expected to adopt an upright posture for the majority of their waking period [8]. Portnoy contributed to this “mechanistic model” by developing an antisiphon device, aiming at the prevention of the effect of siphoning [31,35]. ICP characteristics of siphoning related to postural changes were confirmed in 1990 by Chapman, who utilized a telemetric device in patients with VPS, VAS, and ventriculopleural shunts. Initial investigations centered on the definition of the role of ASD revealed that they were, in general terms, effective in the restoration of “normal pressures” in the upright position [36].

### 3.3. Clinical Manifestations in Shunt Over-Drainage Postural Headache

The clinical manifestations of over-drainage of CSF may be present in an acute pattern, and this complication is not intimately related to the development of chronic refractory headaches [37,38]. A minority of patients may not even manifest any symptomatology after the adoption of low values of intracranial pressure [39]. When a constellation of symptoms appears, they more commonly consist of a “low-pressure headache”, i.e., a headache that is intimately associated with the patient’s posture or “spinal headache”. This is clinically manifested with the patient being unable to assume an upright position. The constellation of symptoms may also include nuchal or upper back pain, nausea, vomiting, dizziness, fatigue, irritability, gait disturbance, diplopia, seizures, and lethargy [40,41]. Symptomatology associated with low intracranial pressure may eventually evolve to intermittent disabling headaches. The next step in the evolution of this clinical syndrome is related to chronic pathological entities, which include developmental delay, decline in school performance, and social withdrawal. When the clinical records of these patients are carefully reviewed, multiple episodes of shunt revisions are frequently registered, which are in accordance with episodes of severe and intractable headaches.

### 3.4. Sequale of Over-Drainage

The conception of excessive drainage of CSF was presented by Dandy in 1932. In 1968, Becker et al. utilized the term “over-drainage” in order to explain the pathophysiologic substrate of the mechanism by which the over-drainage can induce depression of the fontanelle, as well as overriding sutures, craniosynostosis, low ventricular pressure, and, finally, small ventricles [7,8]. This sequence of events involves only the infant population. Pudenz et al. first published a review article centered on over-drainage that was causally related to insertion of a shunt device in 1991 [10]; they concluded that premature closure of cranial sutures and skull deformities (in infants), stenosis or occlusion of the aqueduct, SVS, and low-ICP syndrome are all included in the constellation of manifestations that constitute over-drainage.

In 2018, Ros et al. published a review centered on shunt over-drainage syndrome, attempting to specify the constellation of clinical characteristics that constitute over-drainage. These include headache, with or without associated vomiting and neurological signs or symptoms, plus different degrees of altered consciousness in association with the radiological evidence of small ventricular size and subdural collections of blood or fluid [15]. Current evidence suggests that over-drainage can manifest with a broad spectrum of clinical manifestations; these could include postural headache, subdural hygromas/hematomas, stenosis/occlusion of the aqueduct of Sylvius, craniosynostosis, SVS (characterized by intermittent headache, small ventricles, and slow refilling of the ventricular shunt reservoir), and obstruction of the ventricular catheter. Even though the concept of “over-drainage” has been identified as an adverse effect related to the surgical management of hydrocephalus for several decades, the absence of a strict circumscription, as well as consistent terminology to delineate the concept of over-drainage based on bibliographic reports, is conspicuous. 

### 3.5. Prevention of Headaches Associated with Shunt Over-Drainage: A Brief Summary of Existing Data Regarding Their Pathophysiology, Clinical Symptoms, Treatment, and Prevention

According to a recently published data base [42,43], in about 3% of cases that necessitated a shunt revision procedure, the underlying pathophysiologic mechanism was recorded to be excessive CSF drainage. Nevertheless, the actual relevant rate is rather underestimated, with experts raising this percentage to the rate of 20% of cases. Several techniques aiming towards the reduction of the rate of CSF drainage have been described, incorporating the use of high-pressure non-programmable fixed differential pressure valves, flow valves, and programmable differential pressure valves [11,44,45]. A major drawback that is inherently associated with these cases is related to the fact that CSF drainage may not be as is required when the patient assumes a vertical posture. On the contrary, any attempt to solely increase the differential pressure of the valves was not associated with encouraging results in several published series [11,44,46,47]. These observations forced scientists to develop new mechanistic models related to the pattern of shunt drainage protocols. The main representative of these newly developed drainage systems was included under the umbrella of antisiphon systems. The main aim associated with their development was the prevention of gravitational-related over pull of CSF when the patient is attempting the upright posture. The common concept that underlies the function of antisiphon systems is that they are supposed to be able to adapt to alternating clinical situations or physical conditions, such as the change from the supine to the erect posture [11,48].

Another pathophysiologic mechanism that is inherently related to the development of intractable headaches, especially in the pediatric population, is related to the concept of the slit ventricle syndrome. This is widely recognized as one of the potential side effects of CSF over-drainage, and its pathophysiologic explanation is primarily associated with the acquisition of a pathologically diminished cerebral compliance with a typical leftward shift of the curve in the pressure/volume graph. The collapsed ventricular configuration represents the most typical radiographic feature of SVS. This feature by no means could be considered as been pathognomonic of SVS, as many patients may not exhibit any clinical symptoms. The exact prevalence of a collapsed ventricular system is not universally accepted, although it has been reported in the range of 10–85% of all shunted patients [49]. A wide variety of clinical symptoms have been related to SVS, and a world-wide unanimity regarding its definition does not exist. Nevertheless, classic SVS clinical features consist of severe and persistent or recurrent headaches, frequently related to or provoked by positional changes. The constellation of symptoms, apart from headaches, include vomiting, weakness, ataxia, seizures, cranial nerve deficit, bradycardia, and systemic hypertension, especially in more compromised patients [50,51]. The referring physicians have attempted several positional changes, along with valve upgrade as recommended by several literature reviews [52,53,54,55]. Several patients have undergone repeated procedures aiming toward valve replacement (using valves without any antisiphon system). Nevertheless, none of these interventions have provided permanent relief of the symptomatology of the affected individual. 

Apart from small ventricular size, patients suffering from SVS may present with several indirect radiologic signs of over-drainage. These include a small-sized posterior fossa, hyperostosis of the calvarium, dolichocephalic disproportion, suture sclerosis in proximity to the skull base, parenchymal calcifications, and/or sinus hyperpneumatization [56,57,58]. MRI may prove to be a useful diagnostic modality, as it may offer valuable details about the ventricular and cistern anatomy [52,59,60,61]. Relevant—albeit not usual—MRI findings include the existence of epidural venous plexus engorgement [52,62], along with lumbar canal stenosis [11,63,64]. Other anecdotally reported findings include the existence of pneumocephalus, as well as isolated ventricles [64], along with extra-axial collections of fluid or blood [51,65,66,67]. 

The proposed treatment algorithm for these groups of patients varies greatly [68,69]. Regarding the less-severe cases, the current trend is the selection of a conservative management protocol [52,53,54,55]. In general terms, the management of SVS should aim to restore the pathologically reduced cerebral compliance. Several treatment modalities have been proposed, thus reflecting the inhomogeneity and complexity of the implicated pathophysiologic mechanisms, as no one individual pathogenetic theory could explain the wide variety of clinical manifestations of this syndrome. Other treatment modalities, such as ETV, lumbar drainage, and cranial expansion, have been utilized in refractory cases [70]. Nevertheless, the treatment of SVS may be associated with a wide range of complications and failures to manage it successfully; the exact prevalence of all these complications is unclear as the relevant literature is mainly based on case reports rather than clinical series [51,70,71].

### 3.6. Management of SVS: A Brief Summary of Management Protocols, Including Our Clinical Experience

The most commonly utilized therapeutic measurement, as the first step of our treatment algorithm regarding SVS, is related to upgrading the valve to higher opening pressure values. Although it is technically easy, the overall handling of cases that are managed in such a manner is generally demanding [71,72]. The clinical experience that we have gained with the management of such cases has pointed out the significance of the incremental titration of the valve pressure settings, which means that one level setting adjustment at a time is the only safe and acceptable strategy. According to most centers’ recorded data, this option offers the most effective alleviation of the relevant symptomatology in the greater percentage of individuals who are suffering from a mild range of symptoms. This is especially true for the pediatric cohort of patients, and this seems to be due to the lesser disturbance of the curvature that follows the cerebral compliance as the time course of the disease is sooner and the diagnosis is relatively earlier registered. On the contrary, we have realized that the more severe or more chronic the clinical equivalent of the syndrome, the lesser the chances that a positive and long-lasting response will occur or, more importantly, will be permanent. Reinforcing this view is the fact that it is based on a recent relevant study [42], which enrolled a subgroup of 16 patients that were severely affected and who were improved by valve reprogramming.

Another subgroup of patients has failed these conservative measurements, and it requires surgical treatment. The current trend is to initially attempt externalization of the existent shunt. This therapeutic manipulation offers us the opportunity to obtain valuable and measurable evidence of the initial opening pressure, as well as the possibility to monitor the fluctuations of the ICP values. These variations in the measurements of ICP could be used as guidance when we attempt to increase the reservoir height. There are reports that have proposed that the spontaneous ICP fluctuations, along with the ICP variations to consecutive alterations of reservoir height, could be considered as the mainstay for the invention of the ongoing management options [38]. Several relevant studies [73,74,75,76] have adopted a treatment protocol, which is based upon the ICP values. More precisely, when individuals manifesting with normal or high ICP are being managed, even when the ventricular system is considered to be small, the initial management option was direct shunt replacement using programmable differential pressure valves, which incorporated an antisiphon system. On the other hand, for patients with low ICP measurements, an EVD was the treatment modality of choice. Following that, the ICP gradually increased by a progressive increasement of the reservoir height. For these patients who demonstrated ventricular enlargement concurrently with an increase in ICP values, the proposed management option was an ETV, based on the hypothesis that it could restore the cerebral compliance towards normal values and equilibrate the pressure gradient between ventricles and subarachnoid spaces. There are reports that support the efficacy of this treatment modality [76].

Another subgroup of patients includes those cases which demonstrate a ventricular system whose dimensions remained unchanged, despite the increases in ICP measurements. They received a new programmable differential pressure valve, with an incorporated antisiphon system. There are data which support that upgrading of the valve opening pressure obviated the need for—or at least delayed—surgical intervention in one third of cases. Moreover, according to published studies, an initial attempt based on conservative treatment constitutes a reasonable suggestion [44,69,75,77]. It is widely accepted that young patients’ age and the utilization of an antisiphon system are factors that are associated with a significantly favorable outcome. 

In conclusion, we have mentioned several studies which state that VPS replacement constitutes the optimum therapeutic option for SVS; the simultaneous incorporation of an antisiphon device and valve substitution is strongly recommended [11,44,47,48].

In the subgroup of patients who share a significantly reduced cerebral compliance, it could be beneficial to incorporate a programmable antisiphon system in conjunction with the valvular mechanism. This combination offers the capability of gradually modifying either the ICP or the drainage modalities, or even both of them [11,44,48].

### 3.7. Clinical and Radiologic Outcome

Even though there is a considerable addition of knowledge regarding SVS, as well as technical improvements in the embiomechanics of the shunt systems, the overall natural history of SVS continues to be unpredictable in a vast majority of patients. According to a recently published series [42], no more than half of the participants demonstrated complete resolution of their findings in terms of clinical and radiologic improvement. We would like to mention once more that children were associated with significantly better outcomes than their adult counterparts, and a negative association is established as patient’s age increases.

A major drawback when our therapeutic armamentarium regarding SVS treatment has been considered is related to the fact that most cases have been anecdotally reported, and large case series with previously reporting specific results and treatment outcomes are lacking [15,42]. Current treatment targets are mainly centered on the control of CSF over-drainage and on the improvement of cerebral compliance.

We have concluded that patients suffering from hydrocephalus who have initially been treated with a programmable differential pressure valve were associated with a lesser chance of developing SVS. Nevertheless, it seems that the initial placement of antisiphon systems could not provide any protective effect against the development of SVS.

There is consensus that prompt, proper, and, eventually, a more aggressive treatment may lead to better control of the syndrome in all age-groups.

Another important notice is related to the fact that an immediate and appropriate diagnosis is of inherently significant importance. This is explained by the assumption that a protracted clinical course stands for more protracted periods with negatively impaired quality of life. The current trend stands for the importance of the utilization of valves that offer more options for non-invasive interventions, as well as shunt systems that are an integral part of more sophisticated programmable valves. These options could offer a new therapeutic armamentarium in our attempt to attain the prevention and management of this entity. Apart from that, SVS constitutes a challenging problem, and we have to assume that all treatments modalities have failed in a significant percentage of patients. We hope that ongoing technical innovations will essentially aid in the prevention, diagnosis, and treatment of SVS.

In conclusion, the pathophysiologic entities of shunt over-drainage and slit ventricle syndrome should always be included in our differential diagnostic plan whenever we are confronted with intractable headaches that may resemble the inherent characteristics of migraine in a patient who harbors a ventriculoperitoneal or a ventriculoatrial shunt, especially from infancy. In cases where the diagnostic work-up is unable to underline another pathological substrate, we should maintain the suspicion that our patient could fulfill the diagnostic criteria necessary to be considered under the umbrella of shunt over-drainage and slit ventricle syndrome.

The following table (Table 1) presents a proposed treatment and diagnostic algorithm in cases of chronic headache in pediatric patients suffering from SVS. We would like to clarify that this is an algorithm proposed by the authors based on literature data and their clinical experience.

From our perspective, the overall benefit with this article to the readers is its highlighting the importance of the recognition by the scientific community of the type of headache that is related to shunt over-drainage and slit ventricle syndrome, which is secondary to the surgical management of pediatric hydrocephalus, especially in infants. Excessive CSF drainage following the insertion of a ventricular shunt is a well-known complication that is intimately related to the treatment of hydrocephalus. Nevertheless, the absence of a widely accepted definition in the literature is evident, as well as its consequences. There is no consensus regarding the relevant diagnostic criteria, and, because of this, the exact incidence remains unknown. The overall impact of this uncertainty is reflected in the absence of recommendations dedicated to the prevention, management, and treatment of this condition. Since no consensus has been achieved for several decades, we strongly consider that a definition of OD should not be based upon individual but separate opinions;instead, a significant degree of agreement should be achieved by the majority of OD specialists.

## 4. Conclusions

The overall effect of headache disorders on individual patients, as well as on society itself, is extremely difficult to elucidate with clarity and constitutes a target for public health interventions that is difficult, albeit important, to be achieved [1]. Although there is a widespread disability intimately associated with pediatric headaches, this disorder remains under-diagnosed and, most importantly, under-treated and not appropriately managed. 

Moreover, SVS is intimately related to persistent and difficult-to-manage headaches in the pediatric population [15,16]. The main issue that we have to overcome regarding SVS is that its treatment is currently based primarily on sparse anecdotal reports as, to the best of our knowledge, there are no large cases series published which report results and treatment outcomes based on a widely accepted treatment algorithm. Nowadays, our main goal is centered on attempts to control CSF over-drainage and improve cerebral compliance [11,47]. Nonetheless, SVS remains an intractable problem as its management has proved to be insufficient in a significant percentage of patients, which is a fact that cannot be ignored. A promising technical advancement that will help us to associate the clinical parameters of slit ventricle syndrome with the underlying pathology (reduced cerebral compliance)is the innovation of telemetric systems for ICP measurement [78]. We hope that these devices will offer us the possibility of dynamic ICP monitoring in the near future, thus improving our therapeutic armamentarium in terms of the prevention, diagnosis, and treatment of SVS.

## Figures and Tables

**Table 1 children-11-00596-t001:** Proposed treatment algorithm for the prevention/management of headache in pediatric patients suffering from shunt over-drainage/slit ventricle syndrome.

**General Recommendation:**	
Almost always, even at initial shunt insertion, prefer the use of programmable valves with an integrated anti-siphon device. In every case, we should carefully investigate the possibility of central catheter occlusion as the cause of recent-onset headache. In cases of established slit ventricle syndrome, the ventricular size is hardly expected to be enlarged, as in cases of sudden onset hydrocephalus that do not have as a substrate SVS.	
**First step**	Exclude other non-shunt related causes of headache (i.e., migraine).
**Second step**	In case of a headache compatible with shunt over-drainage, upgrade the opening valve pressure (differential pressure)
**Third step**	If the previous step proves to be inefficient and the patients valve lacks an ASD, insert an ASD in line with the valvular mechanism.
**Fourth step**	Upgrade the ASD pressure (in case it is adjustable) or replace the existing ASD with another with higher opening pressure.
**Fifth step**	Replace the valve with another one with a programmable ASD combined with a programmable valve and adjust/upgrade both of them.
**Finally,** always keep in mind that in cases where slit ventricle syndrome is established, ventricular dimensions are not expected to restore to normal. Our ultimate goal is to avoid/eliminate the incidence of headache and not the normalization of the radiological appearance of the ventricular system.	

## Data Availability

No new data were created or analyzed in this study. Data sharing is not applicable to this article.

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
