# Peer review of "Chronic Pediatric Headache as a Manifestation of Shunt Over-Drainage and Slit Ventricle Syndrome in Patients Harboring a Cerebrospinal Fluid Diversion System: A Narrative Literature Review"

_children, 2024, doi:10.3390/children11050596_

Round 1

Reviewer 1 Report

Comments and Suggestions for Authors

"The abstract and introduction of the article do not provide a clear representation of the content. They focus mainly on headaches in general.

The subheadings are mixed and it is recommended to re-read the content of each section to fully cover the main topic. For example, the passage on the prevention of headaches associated with shunt overdrainage mixes the contents of pathophysiology, clinical symptoms, treatment, and prevention.

The conclusion is very long and should be revised to highlight the main and important takeaways."

Comments on the Quality of English Language

Please recheck for plagiarism, spelling, grammar and spelling issues.

Author Response

Dear Reviewer,

Thank you for your valuable comments.

The following corrections have been performed to our manuscript, as requested.

  1. You have mentioned that ' The abstract and introduction of the article do not provide a clear representation of the content. They focus mainly on headaches in general.' We took into consideration your comment and both sections are modified accordingly.
  2. You have reported that ' The subheadings are mixed and it is recommended to re-read the content of each section to fully cover the main topic'. We have modified accordingly the subheadings, as suggested.
  3. You have commented that 'The conclusion is very long and should be revised to highlight the main and important takeaways'. We have modified our conclusion section accordingly, in order to follow your instructions.
  4. You have reported that ' Please recheck for plagiarism, spelling, grammar and spelling issues.'. All relevant corrections are added in the revised version of our manuscript.

Reviewer 2 Report

Comments and Suggestions for Authors

This is an informal discussion of causes and management strategies of shunt overdrainage in pediatric patients. It is not a systematic review. As the authors explain, there is a lack of systematic reports regarding management. Still, the discussion is of interest for the non-specialist and increases awareness.

Comments on the Quality of English Language

Minor editing is necessary

Author Response

Dear Reviewer,

thank you for your valuable comments.

You have mentioned that' Minor editing is necessary'. We have seriously taken into consideration your comments and all necessary corrections have been included in the revised version of our manuscript.

Reviewer 3 Report

Comments and Suggestions for Authors

Dear Authors,

This is a well-written and informative review suitable for publication in this journal. The text is comprehensive and the data are relevant to clinical practise, describing issues that are difficult to address.

I recommend a minor revision. I have some minor questions or recommendations:

- Introduction: since migraine is not the main topic, the authors can write a word about shunt-related headaches. I have the impression that the introduction is well written but only talks about migraine.

- It is strange that the results are missing. I would prefer the authors to add this section. If the editor agrees, the article can appear without it.

- The discussion is well written. Line 111: better subdural hygromas.

- The conclusions are too long. The authors could remove some parts of the conclusions and move them to the Discussion section.

Author Response

Dear Reviewer,

Thank you for your valuable comments. We have seriously taken into consideration your recommendations and we have performed the following modifications.

  1. You have mentioned that ' Introduction: since migraine is not the main topic, the authors can write a word about shunt-related headaches. I have the impression that the introduction is well written but only talks about migraine.' Some relevant modifications have been added in the revised version of our manuscript.
  2. You have reported that 'It is strange that the results are missing.' We have modified one subheading, explaining that the results are presented in the same section as the 'Management of SVS'.
  3. You have mentioned that 'The discussion is well written. Line 111: better subdural hygromas'. We strongly appreciated your comment and the relevant correction is added to the revised version of our manuscript.
  4. You have commented that 'The conclusions are too long. The authors could remove some parts of the conclusions and move them to the Discussion section'. We strongly appreciate your comment. The conclusion and discussion sections have been modified accordingly.